# Influenza Virus RNA Synthesis and the Innate Immune Response

**DOI:** 10.3390/v13050780

**Published:** 2021-04-28

**Authors:** Sabrina Weis, Aartjan J. W. te Velthuis

**Affiliations:** Division of Virology, Department of Pathology, University of Cambridge, Addenbrooke’s Hospital, Cambridge CB2 0QQ, UK; Sabrina.Weis@stud.uni-heidelberg.de

**Keywords:** influenza A virus, RNA polymerase, RIG-I, mini viral RNA, defective interfering RNA, MAVS, interferon

## Abstract

Infection with influenza A and B viruses results in a mild to severe respiratory tract infection. It is widely accepted that many factors affect the severity of influenza disease, including viral replication, host adaptation, innate immune signalling, pre-existing immunity, and secondary infections. In this review, we will focus on the interplay between influenza virus RNA synthesis and the detection of influenza virus RNA by our innate immune system. Specifically, we will discuss the generation of various RNA species, host pathogen receptors, and host shut-off. In addition, we will also address outstanding questions that currently limit our knowledge of influenza virus replication and host adaption. Understanding the molecular mechanisms underlying these factors is essential for assessing the pandemic potential of future influenza virus outbreaks.

## 1. Introduction

The influenza viruses are negative-sense RNA viruses and members of the family of orthomyxoviruses. Within this family, four influenza virus types are recognized, identified by the letters A, B, C and D. All influenza types can infect vertebrates, including birds, pigs, and aquatic animals, but only the influenza A and B viruses (IAV and IBV, respectively) can cause morbidity and mortality in humans. Influenza C viruses (ICV) can infect humans as well, but they are rarely associated with symptomatic disease in humans, whereas influenza D viruses (IDV) have not caused infections in humans so far. Based on their impact on human health and the extensiveness with which they have been studied, we will here mainly discuss our understanding of IAV infections.

IAV virions consist of a host cell-derived membrane and various viral proteins [1]. On the outside of the membrane, the virion contains the M2 ion channel protein, and the haemagglutinin (HA) and neuraminidase (NA) glycoproteins (Figure 1A). Based on the binding of the HA and NA surface proteins to well-defined antibodies, and more recently, sequencing methods, IAV strains are classified into subtypes. Currently, at least eighteen different HA and eleven different NA classes have been identified [2]. Inside the virion, viral ribonucleoprotein (vRNP) complexes, consisting of viral RNA (vRNA), the RNA polymerase, and nucleoprotein (NP) molecules (Figure 1B), are bound by the viral matrix protein M1. In a complete IAV virion, eight vRNPs are present, containing negative sense vRNA segments ranging from 890 to 2341 nucleotides in length [3]. The 5′ and 3′ termini of each segment are conserved and serve as binding sites for the viral RNA polymerase [4,5]. The NP molecules bind to the phospho-backbone of the vRNA in a sequence-independent manner.

The IAV RNA polymerase catalyses viral replication and transcription in the context of vRNPs [6,7]. Shortly after the infection and import of the vRNPs into the host cell nucleus, the RNA polymerase initiates primary transcription of the vRNA segments. Transcription of a vRNA is a primer-dependent mechanism that involves snatching 5′-capped primers from nascent, capped host polymerase II (Pol II) RNAs [8]. The IAV RNA polymerase also adds a poly-A tail to viral mRNAs, ensuring that they mimic host transcripts [9,10]. After synthesis of new viral RNA polymerase subunits and NP monomers by host cell ribosomes and import of these proteins into the nucleus, the viral RNA polymerase commences replication of the vRNAs via a positive-sense complementary RNA (cRNA) intermediate. The cRNAs also associate with NPs and the viral RNA polymerase to form cRNPs [11]. Using these cRNA intermediates as a template, the RNA polymerase produces more vRNA copies that are assembled into vRNPs and, subsequently, into new virions (Figure 2).

The IAV RNA polymerase produces vRNA copies with a certain nucleotide misincorporation rate [12]. Introduction of base changes in the HA- or NA-encoding vRNA segments contributes to the antigenic drift of IAV strains, enabling them to evade pre-existing immunity. Larger changes in the genetic make-up of the IAV genome occur when reassortment of the genome segments of different IAVs lead to an antigenic shift. When these newly reassorted viruses transmit through a population that is immunologically naïve to the new variant, the new virus can cause a pandemic. In the last century, such IAV pandemics have occurred in 1918, 1957, 1968 and 2009 [13,14,15].

The mechanism underlying influenza disease and the factors that define low and highly pathogenic strains are not completely understood. Current research suggests that a dysregulation of the host immune response, reacting to the effects of the viral infection and releasing cytokines to achieve clearance of the viral infection, plays a key role in pulmonary tissue damage, systemic disease, and death in infections with highly pathogenic strains [16,17,18]. However, the increase in cytokine expression cannot be fully explained by increased replication of an IAV strain in a naïve host [19,20] and the process has been linked to various viral factors, including minor viral proteins, such as PA-X, differences in virus–host interactions, and aberrant viral RNA synthesis [21,22,23]. In this review, we will discuss the role of viral RNA synthesis in triggering the activation of the RIG-I-dependent innate immune response and its contribution to disease.

## 2. The Viral RNA Polymerase and Mechanisms of RNA Synthesis

### 2.1. The Influenza Virus RNA Polymerase

The RNA polymerase is a heterotrimeric enzyme consisting of three subunits: polymerase basic 1 (PB1), PB2 and polymerase acidic (PA). The complete structures of the IAV, IBV and ICV RNA polymerases have been solved through high-resolution X-ray crystallography and cryo-EM (Figure 1C) [24,25,26,27]. The various states that these complexes can adopt have been comprehensively reviewed recently [7,28]. Briefly, the enzyme contains a central core consisting of an RNA-dependent RNA polymerase (RdRp) domain. In addition to the active site, which contains the conserved RdRp motifs, A–F, that are essential for catalysis [6,29], the RdRp domain harbours a β-hairpin, or priming loop, that protrudes into the active site [30]. The priming loop acts as a stacking platform for terminal initiation on the vRNA promoter, and it helps regulate realignment during transcription and replication [30,31,32]. The RNA polymerase core is flanked by several flexible domains that contain other enzymatic functions, e.g., capped RNA binding (CapB) or endonuclease activity (Endo), or host protein binding interfaces, e.g., the PB2 627-domain and nuclear localization signal (NLS) [6,7]. Of these, the 627-domain appears dispensable for the core activity of the RNA polymerase [33], but it is critical for viral replication in cell culture.

Several channels lead to and from the active site: the template entry channel, the NTP entry channel, the product exit channel, and the template exit channel (Figure 1D) [6]. The template entry channel allows entry of the vRNA or cRNA 3′ end into the active site to facilitate initiation. The template exit channel is located close to the entry channel and connected to a 3′ end binding site on the PA subunit surface via a groove on the side of the RdRp domain [9]. On the other side of the RNA polymerase, the product exit channel also performs a key role and allows egress of the replication products as well as entry of a capped RNA primer prior to transcription initiation.

The RNA polymerase binds the 3′ and 5′ ends of the vRNA promoter using binding pockets on the surface of the PB1 and PA subunits [25]. Upon binding, the first 10 nucleotides of the 5′ end of the promoter form a stem-loop structure [24], while the 3′ end is either bound on the surface of the RNA polymerase or transiently tethered through the template entry channel of the RdRp domain [34,35].

### 2.2. RNP Structure

The IAV genome segments are encapsidated in RNPs [36]. In RNPs, NP monomers form a long filament that forms a double helix that is capped by the viral RNA polymerase at one end (Figure 1B) [37]. This RNA polymerase is also referred to as the cis-acting or resident RNA polymerase [7]. Various studies have shown that the NP filament is flexible and can adopt a more relaxed or circular conformation depending on factors such as the length of the bound vRNA and the salt concentration of the surrounding buffer [38,39]. Moreover, inside vRNPs, NP monomers can rotate relative to each other [40]. This rotation can be locked by IAV inhibitor nucleozin, suggesting that the flexibility of the NP monomers in the double helix may be important for IAV RNA synthesis [40].

NP monomers consist of a head and body domain. In an RNP, the monomers are linked via a tail-loop structure that protrudes from the first NP and binds to the insertion groove of the second NP. vRNA or cRNA molecules are bound by a positively charged groove located between the NP head and body domains via the negatively charged RNA backbone [41]. It is likely that, by coating the vRNA and cRNA molecules, the NP filament protects them from degradation and from forming long dsRNA molecules that can be detected by host pathogen receptors. In addition, the NP filament provides a scaffold for the formation of tertiary RNA structures, enabling RNA–RNA interactions between segments [42].

### 2.3. Influenza Virus Transcription

The transcription of the vRNA segments into viral mRNAs with a 5′ cap and a 3′ poly-A tail is performed by the resident viral RNA polymerase. Since the IAV polymerase lacks capping activity, the enzyme carries out a process called ‘cap-snatching’ to generate capped RNA primers that can prime viral mRNA synthesis [43]. Cap-snatching starts with the binding of the IAV RNA polymerase to a serine-5 phosphorylated host Pol II C-terminal domain (CTD) [44]. Serine-5 phosphorylated Pol II complexes are typically initiation complexes that are enriched near transcription start sites, and the serine-5 phosphorylated CTD normally serves as binding site for the host capping machinery and transcription elongation factors. The CTD binding site of the IAV RNA polymerase is located on the outside of the thumb subdomain and involves the PA C-terminal region in the IAV RNA polymerase and residues of PA and PB2 in the IBV and ICV RNA polymerases [8,45,46]. It is likely that other factors, such as CDK9, which phosphorylates the CTD at serine-2 residues and helps Pol II transition from an initiation to an elongation complex, are also involved in recruiting the IAV RdRp to Pol II [47].

Following CTD binding, nascent host RNAs containing a cap-1 structure are bound by the PB2 cap binding domain and cleaved approximately 9–14 nucleotides downstream of the 5′ cap by the PA endonuclease, thus generating primers with a 3′-hydroxyl group [3,48,49]. Interestingly, studies have shown that the IAV RNA polymerase has a bias for different primer length preferences depending on the IAV genome segment it is transcribing [49,50]. By inserting the capped primer via the nascent strand exit channel and adding an NTP to the 3′ end of the cap, transcription is initiated [51]. Depending on the stability of the primer–template duplex, which is affected by the length and sequence of the primer, the partially extended capped primer can be fully extended, or first realigned to the 3′ end of the vRNA prior to processive transcription elongation, duplicating the ends of the genome as a result [32,49,52]. It is possible that this realignment mechanism evolved to increase the likelihood of successful viral transcription events with suboptimal primers, e.g., primers that are too short or that do not have the 3′ terminal sequence to stably base pair with the template [32]. The cap-snatching mechanism also enables the polymerase to express alternative open-reading frames, using start codons present in the capped RNA primer [53].

Transcription elongation takes place until a sequence of 5–7 U-residues, located approximately 16–17 nucleotides from the 5′ end of the vRNA template, triggers transcription termination and polyadenylation [9,10]. The poly-A tail is generated by polymerase stuttering on the U-stretch due to steric hindrance because the 5′ end of the vRNA template remains bound to the promoter binding site of the transcribing polymerase [9,10]. When the 5′ cap of the viral mRNA is released by the RdRp, it binds to the nuclear cap binding complex, which recruits cellular factors to facilitate assembly of viral mRNAs into messenger ribonucleoprotein complexes (mRNPs) [54]. This strategy generates viral transcripts that are structurally indistinguishable from cellular transcripts for the host immune system, enabling mRNA nuclear export and translation of viral proteins by cellular mechanisms. Meanwhile, the template takes a different path from the nascent transcript and leaves the active site via the template exit channel and a groove on the outside of the thumb subdomain [9]. The 3′ terminus of the template binds on the outside of the PA subunit, also referred to as the B-site, and will likely return to the active site once polyadenylation has been completed and the nascent strand is released [9].

### 2.4. Influenza Virus Replication

By contrast to IAV transcription, IAV replication is a two-stage process that is initiated in a primer-independent manner. During the first step of replication, cRNA synthesis takes place by de novo initiation on the 3′ terminus of a vRNA segment [55]. The RdRp domain uses the priming loop to stabilize the pppApG initiation product [30]. Recent structural evidence has also revealed that host protein ANP32A must bind to the RNA polymerase residing in the vRNP in order to recruit a newly synthesized RNA polymerase [56]. This process leads to the formation of a viral RNA polymerase dimer that likely facilitates the transfer of the emerging 5′ terminus of the nascent strand to the new RNA polymerase and the formation of a cRNP [11,56,57]. No polyadenylation takes place during cRNA synthesis and elongation continues until the vRNA template has been fully copied, which implies that the 5′ end of the vRNA needs to be released from its binding pocket during cRNA sythesis. The mechanism that triggers this release has not been identified so far. 

The cRNA 5′ and 3′ ends are thought to bind to the same binding sites as the vRNA terminal sequences [58,59], although some recent structures have shown that an RNA oligo representing the cRNA 3′ terminus can bind to the B-site in the absence of elongation [60,61]. However, sequence differences between the vRNA and cRNA promoters mean that the RNA polymerase has different binding constants for the two promoters, with a higher binding constant for the vRNA promoter [35]. To initiate vRNA synthesis from the cRNA template, a pppApG dinucleotide is synthesized internally on residues 4 and 5 of the cRNA 3′ end and subsequently realigned to the cRNA 3′ terminus to make a full-length copy of the cRNAs [55]. Recent studies showed that the priming loop and the binding of a regulatory polymerase, likely the previously identified trans-activating polymerase, play an important role in controlling this realignment event [11,31,60,62]. Furthermore, a third polymerase must bind near the product exit channel of the resident RNA polymerase in order to form the new vRNP complex, similar to the mechanism observed for cRNP assembly [7,56,57,62].

### 2.5. Aberrant RNA Synthesis

In addition to full-length vRNA and cRNA molecules, the IAV RNA polymerase can produce RNA products that are shorter than the vRNA or cRNA template from which they derive. These aberrant RNA species include small viral RNAs (svRNA) [63], defective interfering RNAs (DI RNA) [64], and mini viral RNAs (mvRNA) [21]. svRNAs represent the 5′ triphosphorylated 22–27 terminal nucleotides of vRNAs [63], whereas DI RNAs and mvRNAs are, respectively, >180 nt and 56–125 nt long viral genome segments that contain internal deletions between the conserved 5′ and 3′ termini (Figure 3). In addition to their varying size, a key difference between DI RNAs and mvRNAs is that mvRNAs can be replicated independent of NP [21,65], suggesting that they may not reside in typical IAV vRNPs. Recent studies have demonstrated that IAV aberrant RNAs can be found in infected mice, ferrets or humans [21,66,67], and that there is a link between DI RNA or mvRNA production and IFN induction [21,68]. Such a link has not been observed for svRNA synthesis [63].

It is currently not fully understood what triggers IAV aberrant synthesis. In tissue culture experiments, IFN-inducing mutations have been linked to the accumulation of high numbers of DI genomes. For instance, Perez-Cidoncha et al. performed selection experiments in non-IFN-responsive cells to isolate virus mutants that were unable to prevent IFN induction. The identified mutants were shown to be stronger IFN inducers and to generate higher numbers of DI genomes than wild-type viruses [69]. Similarly, next-generation sequencing and mutational analyses of influenza viruses capable of inducing IFN promoter activation revealed influenza virus RNA polymerase mutants that have the potential to increase aberrant RNA synthesis [70,71]. Moreover, it was recently found that RNA polymerase variants associated with avian-adapted mutations produce more mvRNAs in human cells [21]. Based on these observations, it was speculated that some of these RNA polymerase mutants do not function optimally and generate aberrant RNA products, potentially contributing to the activation of the host innate immune response in infections with highly pathogenic IAVs [21].

Another trigger for aberrant RNA synthesis and, in particular, mvRNA synthesis appears to be limiting NP levels, as shown by experiments that either overexpressed the viral RNA polymerase subunits or knocked down NP expression [21,72,73]. This suggests that an incompletely encapsidated RNA template (aberrant RNP) or an interrupted encapsidation process (limiting free NP) may lead to impaired RNA polymerase processivity and/or a higher propensity to realign product and template, and the generation of an aberrant RNA. It also suggests that some antivirals that target NP and reduce RNA polymerase processivity may trigger aberrant RNA synthesis, but this has not been studied in detail [40].

It is currently not fully understood how IAV aberrant RNAs are synthesized, though it has been ruled out that splicing or ligation mechanisms are involved [74]. Deep-sequencing analyses suggest that a combination of copy-choice recombination and poor RNA polymerase fidelity play some role in the process [21,67,75]. However, DI RNA and mvRNA sequences have been identified that were likely not generated through a copy-choice mechanism [21,76], suggesting that other factors may contribute to the synthesis of DI RNAs and mvRNAs. Additional studies have suggested that A/U-rich sequences may be involved [77], but these observations were not supported by analyses from other groups [76,78]. Additional studies in this area are evidently needed to better understand how the synthesis of aberrant RNAs is executed by the RNA polymerase.

## 3. Host Pathogen Receptors, Detection of Viral RNAs, and Host Shut-Off

The first line of defence during an infection relies on the recognition of conserved structures exhibited by pathogens, known as pathogen-associated molecular patterns (PAMPs). These PAMPs can be sensed by host proteins known as pattern-recognition receptors (PRRs). IAV is recognized by at least three distinct families of PRRs, including toll-like receptors (TLRs), the nucleotide oligomerization domain (NOD)-like receptors (NLRs) and RIG-I-like receptors (RLRs) [79]. These PRRs facilitate viral sensing in distinct cellular compartments of different cell types and at different infection phases. TLR7, for instance, recognizes incoming virions in endosomal compartments of plasmacytoid dendritic cells (pDCs) and induces type I interferons (IFN) production via myeloid differentiation primary response 88 (MyD88) [80]. Another example is TLR3, which is constitutively expressed in macrophage endosomes, and helps detect infected cells and initiate pro-inflammatory responses through TIR-domain-containing adapter-inducing interferon-β (TRIF) signalling [79,81]. One well-known member of the NLR family responding to IAV infection is the NLRP3 signalosome, which is resident in the cytosol of various cell types and able to trigger interleukin-1β induction [82]. Here, we will focus on RLR member RIG-I, which is the key IAV RNA sensor and IFN inducer in most cell types targeted by IAV, and which has been intensively studied in recent years.

### 3.1. RIG-I, MAVS and the Interferon Signalling Cascade

RIG-I is an RNA helicase that provides viral sensing in the host cell cytosol and nucleus [83,84]. RIG-I consists of a central helicase/ATPase domain, two N-terminal caspase recruitment domains (CARDs) and a C-terminal RNA-binding domain. RIG-I can be activated by a number of RNA ligands, including dsRNAs with a 5′ triphosphate (see below) [85,86]. When RIG-I binds a ligand, it hydrolyses ATP and undergoes a conformational change that exposes the two CARD domains [87]. Exposure of the CARD domains leads to lysine 63-linked ubiquitination by regulatory ubiquitin ligase tripartite motif 25 (TRIM25) and interactions between RIG-I and the mitochondrial antiviral signalling protein (MAVS) (Figure 4) [88,89].

MAVS comprises an N-terminal CARD domain, a proline-rich region and a C-terminal transmembrane (TM) domain. With its TM segment, MAVS is targeted to the outer mitochondrial membrane, a location that is crucial for competent signal transduction [90], while the MAVS’ CARD domain facilitates the interaction with the CARD domains of activated RIG-I. The binding of ubiquitinated RIG-I to MAVS leads to activation of the MAVS signalosome [89]. The MAVS signalosome includes members of the TRAF (TNF receptor-associated factor) family, TRADD (TNFR1-associated death domain protein), STING (stimulator of interferon genes) protein families, and molecules to recruit TBK1 (TANK-binding kinase 1), which is required for the phosphorylation of interferon regulatory factor 3 (IRF3) and IRF7. The latter two proteins ultimately stimulate the expression of type 1 interferon (IFN) genes (Figure 4) [89]. Besides IRF3/7 signalling, the MAVS signalosome induces the nuclear factor-kB (NF-kB) pathway by activating the IkB kinase (IKK) complex, consisting of the catalytic subunits IKKα and IKKβ and the regulatory subunit IKKγ. The IKK complex, in turn, phosphorylates the NF-kB inhibitor α (IkBα) to target the inhibitor for proteasomal degradation. Consequently, NF-kB can be released and translocated to the nucleus to initiate expression of pro-inflammatory cytokine genes (Figure 4) [89,90].

MAVS proteins have been shown to self-associate via their TM domains to create a platform for the interaction with downstream signalling proteins [91]. Furthermore, mitochondrial dynamics have been proposed to play an important role in modulating downstream signalling and it has been reported that the fusion of mitochondria, a process that is mediated by mitofusin 1 (MFN1) and MFN2, is essential for efficient signal transduction [92]. An intact mitochondrial membrane potential also seems to be required for the formation of MAVS complexes, and a mitofusin deficiency or a decrease in the membrane potential results in a fragmented mitochondrial network and a disruption of the MAVS signalling cascade [93]. Interestingly, several studies have shown that the PB2 subunit of the IAV RNA polymerase can translocate to the mitochondria [94,95], while alternative translation products of the PB1 gene, PB1-F1, also affect mitochondrial function and MAVS signalling [96,97,98], suggesting that IAV has evolved several polymerase segment-encoded mechanisms to disrupt RIG-I signalling via MAVS. Additional viral proteins that modulate the innate immune response are NS1 and PA-X, whose roles have been extensively discussed elsewhere (Figure 4) [99].

### 3.2. Z-DNA Binding Protein 1 and Necrosis

In addition to RIG-I sensing, recent research has shown that Z-DNA binding domain protein 1 (ZBP1 or DAI) is able to bind IAV RNAs [100,101]. After IAV RNA binding, ZBP1 triggers RIPK3-mediated mixed lineage kinase domain-like pseudokinase (MLKL)-dependent envelope disruption, leakage of DNA into the cytosol, and eventual necroptosis. Like RIG-I, ZBP1 appears to preferentially bind shorter IAV RNAs and aberrant RNAs. Why ZBP1, which was originally identified to bind left-handed DNA helices or Z-form DNA, binds these RNAs and why these IAV RNAs molecules have Z-form dsRNA regions is not understood [101].

### 3.3. Detection of Influenza Virus RNA

Various in vitro studies have been performed to identify the RNA species that bind and activate RIG-I. Most of these RIG-I agonists contain a short double-stranded RNA sequence and a 5′ terminal triphosphate (ppp) group [102], but other RNAs, such as RNAs with a 5′ diphosphate, can also be bound by RIG-I [83]. A dsRNA length of 10 bp located directly next to the 5′ ppp or a small RNA hairpin with 5′ ppp gives the strongest RIG-I activation [103,104,105,106]. Other features, such as the RNA length, presence of overhangs, and nucleotide composition all appear to play a role in RIG-I activation, as well [86,107,108]. 

The IAV 5′ and 3′ termini form a panhandle structure in solution that consists of partially double-stranded RNA and a 5′ ppp end. This panhandle can act as RIG-I inducer, since in vitro transcripts of influenza virus segments are capable of inducing IFN production when transfected into cells [108], although it must be noted that T7 run-off transcripts often contain double-stranded RNA by-products that also induce IFN expression. Similar transcripts can also activate purified RIG-I in vitro [21,109]. To identify IAV RNA-derived RIG-I agonists in their physiological environment, Rehwinkel et al. isolated RIG-I complexes from infected cells. Immuno-precipitates contained vRNAs and cRNAs that could be detected by primer extension, suggesting that viral genome segments bearing a 5′ ppp are a natural source of RIG-I induction [110]. In a contemporary study, Baum et al. used next-generation RNA sequencing to demonstrate that RIG-I preferentially associates with short viral RNAs, such as DI RNAs, based on the enrichment for short reads covering the 5′ and 3′ termini of the viral RNA segments [108]. Interestingly, vRNPs have also been observed to interact with RIG-I at very early time points during infection [111], but no innate immune signalling was triggered by this interaction.

While the above studies demonstrate that RIG-I can bind viral RNAs, they do not provide information on the timing and location of RIG-I binding during an infection. Interestingly, no RIG-I ligands could be isolated in infected cells treated with a translation inhibitor [112]. Blocking viral mRNA translation stops the synthesis of new polymerase subunits and NP molecules, and the lack of these viral proteins likely prevents activation of subsequent replication, which is normally facilitated by the regulatory and encapsidating RNA polymerase. Similarly, Killip et al. could limit the effect of IFN stimulation by inhibiting cellular transcription, and thereby, limit that of viral transcription and replication [112], while Rehwinkel et al. showed that IFN stimulation of infected cells is negated by a replication-defective influenza A virus RNA polymerase [110]. Together, these findings suggest that the activation of RIG-I during influenza virus infection requires active viral replication [112,113].

vRNPs and cRNPs protect the viral RNA from RIG-I, in part because the 5′ and 3′ termini are tightly bound by the viral RNA polymerase [24]. If RIG-I recognised PAMPs in the context of vRNPs, dissociation or active removal of the viral polymerase, and possibly NPs, needs to occur to allow RIG-I to bind the viral panhandle. Theoretically, this could happen when cells are infected with a high MOI, when RNA polymerase mutations reduce Pol II binding or cap-snatching, when mutations increase viral replication, or when viral RNA polymerase or NP levels are not sufficient to complete encapsidation of nascent vRNA or cRNAs. Interestingly, recent studies have indeed shown that limiting NP levels leads to the formation of aberrant viral RNAs and RIG-I activation during IAV infection as well as other negative-sense RNA virus infections [21,72,73]. Moreover, in vivo studies by Tapia et al. confirmed that DI-rich influenza viruses are associated with stronger IFN expression in lungs of infected mice [114]. Interestingly, in a recent comparison among full-length vRNAs, DI RNAs and mvRNAs, mvRNAs were found to be the most potent inducers of IFN promoter activity [21]. It is presently unclear why these short viral molecules are more potent than longer IAV RNAs, but it is tempting to speculate that, because mvRNAs do not need NP for their amplification and may not form (stable) RNPs, they are more easily detected by RIG-I. It has been speculated that aberrant RNAs, in particular DI RNAs, can interfere with the expression of viral antagonists, thus frustrating the virus’ ability to modulate and block the activation of IFN production [113]. Further studies are evidently needed to explore these hypotheses or find alternative explanations.

### 3.4. Host Shut-Off

IAV infections lead to the suppression of host transcription and translation, a process called host shut-off. It has been proposed that host shut-off favours translation of viral mRNAs over host mRNAs, and that it reduces the expression of innate immune response genes. Various mechanisms have been identified that contribute to host shut-off during IAV infections [115], including the expression of viral proteins PA-X and NS1 (Figure 4) [116,117]. Influenza virus cap-snatching also contributes to host shut-off. IAV cap-snatching targets a large variety of Pol II transcripts, including small nucleolar RNAs and small nuclear RNAs [49,50,118]. It is plausible that the IAV RNA polymerase targets Pol II transcripts in an opportunistic fashion, leading to an enrichment of capped primers derived from abundant host transcripts [118]. The impact of these cap-snatching events is a loss of Pol II occupancy on host genes, likely through Pol II transcription termination or inhibition [119].

### 3.5. Detection of Host RNA

In addition to detecting viral RNA by distinguishing it from host RNA, RIG-I can also bind cellular 5S ribosomal RNA pseudogene 141 (RNA5SP141) transcripts during viral infections, including infections with an influenza virus [120]. RNA5SP141 transcripts are generated by Pol III and normally bound by RNA5SP-interacting proteins, precluding detection by RIG-I. During a viral infection, the synthesis of these RNA5SP-interacting proteins is downregulated through host shut-off mechanisms, allowing RIG-I detection of RNA5SP and initiation of the innate immune response [120].

## 4. Conclusions and Outlook

The pathogenesis of IAV infections is a complex process that involves multiple cellular, viral and, occasionally, bacterial factors. Exposure to IAV can lead to a wide range of disease outcomes, and the host innate immune response is here a mixed blessing: it is indispensable for restricting viral replication and spread, but an overstimulation of the inflammatory response may also lead to severe disease. In efforts to explain pathogenic differences between IAV strains, mutations in IAV proteins have been identified that alter the receptor binding specificity, pH sensitivity, or function of IAV accessory proteins, such as NS1, PA-X and PB1-F2. These accessory proteins play a role in modulating the host response upon IAV infection. Mutants derived from avian-adapted IAV strains may be less efficient at modulating the innate immune response in humans than mammalian-adapted IAV strains. IAV proteins, or IAV infections, have also been found to induce alterations in the infected host cell, such as H3K79 acetylation, the release of nuclear RNAs, or Pol II read-through. How all these factors and modulations come together and add up to differences in virulence or pathogenicity is not fully understood. 

In addition to mutations in the viral proteins, it has been discovered that aberrant IAV RNAs produced by erroneous polymerase activity can induce RIG-I activation and the expression of pro-inflammatory cytokines [21,113]. Many questions remain regarding this role of aberrant IAV RNAs. For instance, it is presently unclear why avian-adapted IAV RNA polymerases appear to produce higher levels of mvRNAs than mammalian-adapted IAV RNA polymerases. In addition, it has not been explored if different sequences within aberrant RNA populations contribute equally to RIG-I activation or if sequence-dependent RIG-I activation differences exist among different aberrant RNAs. It is tempting to speculate, for instance, that IAV normally generates low levels of some aberrant molecules that induce no RIG-I activation, so they can act as RIG-I traps. Other work suggests that ZBP1 may also bind aberrant viral RNAs and induce necrosis [100,101]. It is unclear at present if ZBP1 can only bind a sub-population of these aberrant RNAs since it only recognizes Z-form dsRNA, or if all IAV RNA species contain Z-form dsRNA and contribute to ZBP1 signalling. 

In conclusion, the induction of a cytokine storm that underlies the pathogenesis of some human IAV infections may be the result of a combination of a strong RIG-I activation and the inability of the virus to inhibit or evade key players of the host defence system. Indeed, passaging of wild-type IAV in non-IFN-responsive cells results in IAV mutants containing IFN-inducing mutations at conserved positions in the IAV genome, and not just in known IFN-linked sites in, for example, the NS1 gene. This view suggests that the entire IAV genome is optimized for both viral replication as well as counteracting or evading the host response. From that assumption it follows that due to suboptimal interactions with mammalian host factors, avian-adapted IAV proteins may be more likely to generate aberrant RNA products and be less efficient at suppressing the innate immune response. Future studies will be needed to better understand these interactions and their contribution to IAV virulence and pathogenicity.

## Figures and Tables

**Figure 1 viruses-13-00780-f001:**
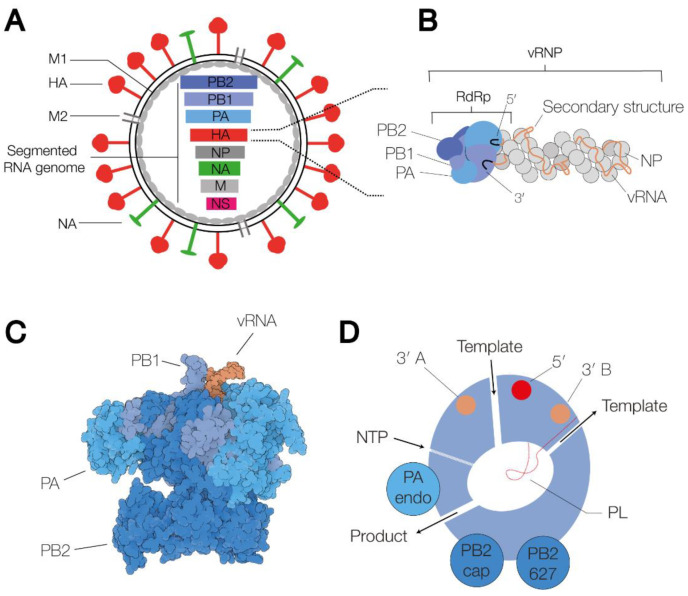
Schematic of the IAV virion, the viral ribonucleoprotein complex, and the viral RNA-dependent RNA polymerase. (**A**) Schematic of the IAV virion. The virion consists of a host cell-derived membrane, the surface glycoproteins HA and NA, the M2 ion channel, the M1 protein capsid, and the viral ribonucleoprotein (vRNP) complexes. The vRNPs each contain a segment of single-stranded, negative-sense viral RNA (vRNA). Eight of these vRNA segments make up the viral genome. (**B**) Schematic representation of a vRNP. The 5′ and 3′ termini of the vRNA (black) are bound by the heterotrimeric RdRp, which consists of the proteins polymerase basic 1 (PB1) (bright blue), PB2 (dark blue) and polymerase acidic (PA) (grey blue). The rest of the vRNA is associated with nucleoprotein (NP) monomers, forming an antiparallel double helix with a closing NP loop. (**C**) Surface model of the IAV RdRp in the transcription pre-initiation state (PDB 4WSB). The three polymerase subunits, PB1, PB2 and PA, are coloured bright blue, dark blue and grey blue, respectively. The 5′ and the 3′ end of the vRNA are coloured orange. (**D**) Schematic of the IAV RNA polymerase, in which the channels and RNA binding sites are indicated. The binding sites of the 5′ and the 3′ end are coloured red and orange, respectively.

**Figure 2 viruses-13-00780-f002:**
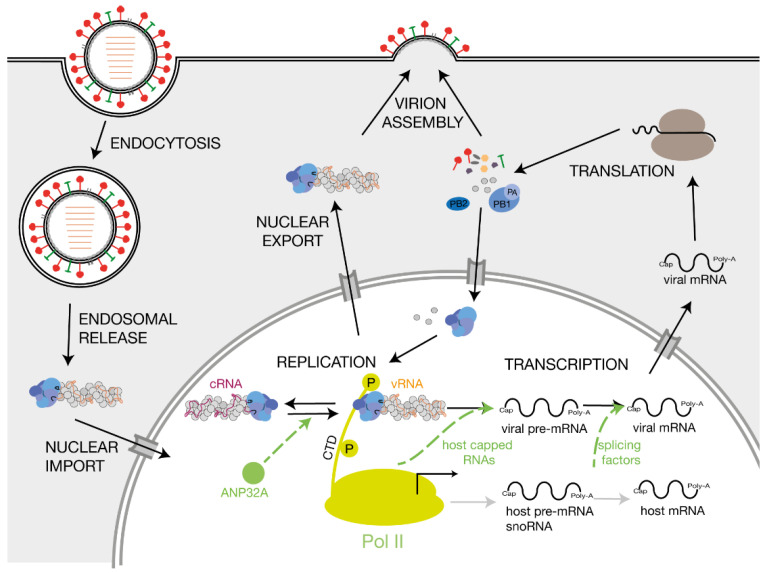
Influenza A virus replication and host factors involved in influenza A virus RNA synthesis. Schematic of IAV replication cycle. Key host factors that play a role in viral RNA synthesis are indicated in green. The green arrows point to the step of the replication cycle that is promoted by the corresponding host factor. The viral proteins PB1, PB2 and polymerase acidic PA are coloured bright blue, dark blue and grey blue, respectively.

**Figure 3 viruses-13-00780-f003:**
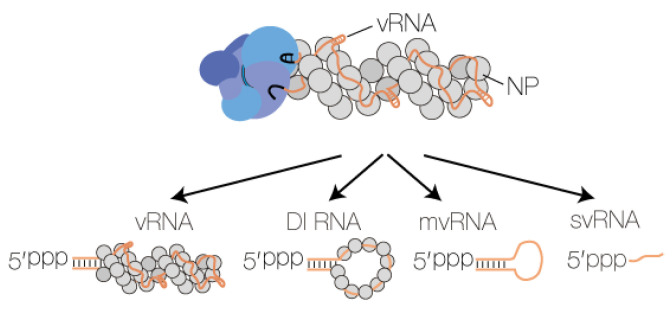
Influenza A virus RNA polymerases produces four types of RNA. The IAV RNA polymerase produces four types of RNA: full length vRNA or cRNA segments, DI RNAs, mvRNAs, and svRNAs. Three of these RNA species contain partially dsRNA sequences and a 5′ triphosphate. When this dsRNA element is exposed to the solvent, it can be bound by RIG-I. The viral proteins PB1, PB2 and PA are coloured bright blue, dark blue and grey blue, respectively.

**Figure 4 viruses-13-00780-f004:**
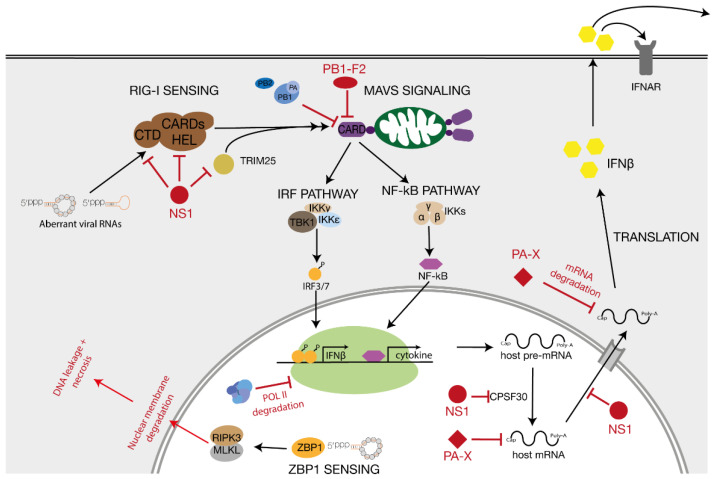
RIG-I and ZBP1 signalling after detection of influenza A virus RNA. After binding to aberrant IAV RNA molecules, RIG-I induces innate immune signalling via the MAVS signalling cascade. MAVS activates the IRF and the NF-κB pathway which lead to the production of IFNβ and inflammatory cytokines, respectively. ZBP1 binds viral RNA in the nucleus and triggers RIPK3 mediated necrosis. Viral antagonists PA-X, NS1 and PB1-F2 that block certain steps of the immune signalling pathways are also indicated.

## Data Availability

Not applicable.

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
