# Peer review of "Influenza Virus RNA Synthesis and the Innate Immune Response"

_viruses, 2021, doi:10.3390/v13050780_

Round 1

Reviewer 1 Report

The review article by Weis and te Velthuis provides an overview about the interplay between RNA synthesis and innate immunity for influenza virus. Overall, the review is comprehensive and well-written. In the abstract, the authors mention that understanding the molecular mechanisms for viral replication and host response are important for developing antiviral strategies, it is not clear however from the review how the discussed factors can be used to tailor specific antiviral intervention strategies against this group of viruses.

Author Response

The review article by Weis and te Velthuis provides an overview about the interplay between RNA synthesis and innate immunity for influenza virus. Overall, the review is comprehensive and well-written. In the abstract, the authors mention that understanding the molecular mechanisms for viral replication and host response are important for developing antiviral strategies, it is not clear however from the review how the discussed factors can be used to tailor specific antiviral intervention strategies against this group of viruses.

We thank the reviewer for their positive comments and apologise for the error in the abstract. We have corrected the abstract in this revised version and removed the point about the antiviral strategies, which we decided not to cover in the main text due to limited existing literature. The revised abstract now reads at line 18-19:

“Understanding the molecular mechanisms underlying these factors is essential for assessing the pandemic potential of future influenza virus outbreaks.”

Reviewer 2 Report

This review article by Weis and te Velthuis focuses on the interplay between influenza virus RNA synthesis and the detection of influenza virus RNA by host innate immune system. It is built upon key and up-to-date literature that describes the generation of various RNA species, host pathogen receptors, and host shut-off. The authors point out that understanding the molecular mechanisms underlying these factors is essential for developing anti-influenza strategies and assessing the pandemic potential of future influenza outbreaks.

Overall, the review is nicely written, clear and largely reflects our current knowledge in the field. Here are, however, a number of issues that should be addressed to improve the manuscript:

Major concerns:

  1. Figure 1. The description of color is not accurate. In figure 1B, the color of vRNA looks like copper not red. The color of PB1 looks purple instead of green. The color of PA looks light blue instead of yellow. In figure 1C, it appears that the three polymerase subunits, PA, PB2 and PB1, are coloured bright blue, dark blue and grey blue, respectively. Not in the order of PB1, PB2 and PA.
  2. It is well-known that there are three important pattern recognition receptors (PRRs) that recognize virus replications and initiate the innate immune responses. They are RIG-like receptors (RLRs), Toll-like receptors (TLRs), and Nod-like receptors (NLRs). Other PRRs, such as TLR3 and TLR7, need to be discussed in the article. Otherwise, consider changing the title of the review article to “Influenza virus RNA synthesis and the RIG-I initiated innate immune response”.
  3. Line 258-268, the authors described the RIG-I/MAVS signaling pathway, where they missed downstream NF-kB pathway. The MAVS signalosome activates the NF-kB signaling pathway besides transcription factors IRF3/7. The pathway is shown in Figure 4 by the authors. Please add it in the main text.

Author Response

This review article by Weis and te Velthuis focuses on the interplay between influenza virus RNA synthesis and the detection of influenza virus RNA by host innate immune system. It is built upon key and up-to-date literature that describes the generation of various RNA species, host pathogen receptors, and host shut-off. The authors point out that understanding the molecular mechanisms underlying these factors is essential for developing anti-influenza strategies and assessing the pandemic potential of future influenza outbreaks.

Overall, the review is nicely written, clear and largely reflects our current knowledge in the field. Here are, however, a number of issues that should be addressed to improve the manuscript:

We thank the reviewer for their positive comments and have done our best to address the issues raised by the reviewer.

Major concerns:

  1. Figure 1. The description of color is not accurate. In figure 1B, the color of vRNA looks like copper not red. The color of PB1 looks purple instead of green. The color of PA looks light blue instead of yellow. In figure 1C, it appears that the three polymerase subunits, PA, PB2 and PB1, are coloured bright blue, dark blue and grey blue, respectively. Not in the order of PB1, PB2 and PA.

We apologize for the error in the legend and have corrected the color description in the figure 1 legend. The new legend for panels C-D now reads (line 700-707):

B) Schematic representation of a vRNP. The 5′ and 3′ termini of the vRNA (black) are bound by the heterotrimeric RdRp, which consists of the proteins polymerase basic 1 (PB1) (bright blue), PB2 (dark blue) and polymerase acidic (PA) (grey blue). The rest of the vRNA is associated with nucleoprotein (NP) monomers, forming an antiparallel double helix with a closing NP loop. C) Surface model of the IAV RdRp in the transcription pre-initiation state (PDB 4WSB). The three polymerase subunits, PB1, PB2 and PA, are coloured bright blue, dark blue and grey blue, respectively. The 5′ and the 3′ end of the vRNA are coloured orange. D) Schematic of the IAV RNA polymerase, in which the channels and RNA binding sites are indicated. The binding sites of the 5′ and the 3′ end are coloured red and orange, respectively.”

  1. It is well-known that there are three important pattern recognition receptors (PRRs) that recognize virus replications and initiate the innate immune responses. They are RIG-like receptors (RLRs), Toll-like receptors (TLRs), and Nod-like receptors (NLRs). Other PRRs, such as TLR3 and TLR7, need to be discussed in the article. Otherwise, consider changing the title of the review article to “Influenza virus RNA synthesis and the RIG-I initiated innate immune response”.

We apologise for not sufficiently describing the role of other pattern recognition receptors in our review. We have now added a new paragraph that better introduces the innate immune section and explains why we focus on the RIG-I pathway for the purpose of our review. This section is added to lines 249-265, and reads:

“The first line of defence during an infection relies on the recognition of conserved structures exhibited by pathogens, known as the pathogen-associated molecular patterns (PAMPs). These PAMPs can be sensed by host proteins known as the pattern-recognition receptors (PRRs). IAV is recognized by at least three distinct families of PRRs, including toll-like receptors (TLRs), the nucleotide oligomerization domain (NOD)-like receptors (NLRs) and RIG-I-like receptors (RLRs) [82]. These PRRs facilitate viral sensing in distinct cellular compartments of different cell types and at different infection phases. TLR7, for instance, recognizes incoming virions in endosomal compartments of plasmacytoid dendritic cells (pDCs) and induces type I interferons (IFN) production via myeloid differentiation primary response 88 (MyD88) [83]. Another example is TLR3, which is constitutively expressed in macrophage endosomes, detecting infected cells and initiating proinflammatory responses through TIR-domain-containing adapter inducing interferon-β (TRIF) signalling [82, 84]. One well-known member of the NLR family responding to IAV infection is the NLRP3 signalosome, which is resident in the cytosol of various cell types and able to trigger interleukin-1β induction [85]. Here, we will focus on the third group of PRRs, the RLRs and especially RIG-I, which is the key IAV RNA sensor and IFN inducer in most cell types targeted by IAV, and which has been intensively studied in recent years.”

  1. Line 258-268, the authors described the RIG-I/MAVS signaling pathway, where they missed downstream NF-kB pathway. The MAVS signalosome activates the NF-kB signaling pathway besides transcription factors IRF3/7. The pathway is shown in Figure 4 by the authors. Please add it in the main text.

We have added a description of the NF-kB pathway to the main text at lines 287-293 and cited Fig. 4 accordingly. This section reads:

“Besides IRF3/7 signalling, the MAVS signalosome induces the nuclear factor-kB (NF-kB) pathway by activating the IkB kinase (IKK) complex consisting of the catalytic subunits IKKα and IKKβ and the regulatory subunit IKKγ. The IKK complex, in turn, phosphorylates the NF-kB inhibitor α (IkBα) to target the inhibitor for proteasomal degradation. Consequently, NF-kB can be released and translocated to the nucleus to initiate expression of pro-inflammatory cytokine genes (Fig. 4) [92, 93].”

Reviewer 3 Report

The review by Weis and te Velthuis examines the detailed mechanisms how influenza A virus synthesis evades or triggers the innate immune response.  This is a very focused review on the viral transcriptional machinery, and builds off the earlier research report by te Velthuis demonstrating that miniviral RNAs are largely detected by RIG-I in the process of IAV replication from Nature Microbiology.  Overall the paper is tightly written and well referenced.  I learned alot from reading this paper.   I have a few minor comments intended to improve the presentation.

In the abstract, the authors indicate that they intend to address outstanding questions that limit our understanding of replication and host adaptation.  I didnt see a discrete section for this, but think it might be worth including a paragraph on these big questions.  For example, the mechanisms how cap snatching would lead to host shut off,  why mvRNA production is more prominent in avian adapted virus come to this readers mind.  The functional role of NS1 in suppressing activated RIG-I is not mentioned, nor why IAV triggers H3K79 acetylation and how this might be related to host shut-off. Reviews should help point the field to major unresolved questions.  

Some other minor points

the sentence in line 388 beginning "Indeed.." should be broken up or modified for clarity.

Fig 1B, the RNA strand is difficult to see in this color schema.

Fig. 2 legend should indicate that viral polymerase components are colored blue.

Fig. 3 legend, line 681-682 needs to be reworked.

Author Response

The review by Weis and te Velthuis examines the detailed mechanisms how influenza A virus synthesis evades or triggers the innate immune response.  This is a very focused review on the viral transcriptional machinery, and builds off the earlier research report by te Velthuis demonstrating that miniviral RNAs are largely detected by RIG-I in the process of IAV replication from Nature Microbiology.  Overall the paper is tightly written and well referenced.  I learned alot from reading this paper.   I have a few minor comments intended to improve the presentation.

We thank the reviewer for their positive comments and have done our best to address the issues raised by the reviewer.

In the abstract, the authors indicate that they intend to address outstanding questions that limit our understanding of replication and host adaptation.  I didnt see a discrete section for this, but think it might be worth including a paragraph on these big questions.  For example, the mechanisms how cap snatching would lead to host shut off,  why mvRNA production is more prominent in avian adapted virus come to this readers mind.  The functional role of NS1 in suppressing activated RIG-I is not mentioned, nor why IAV triggers H3K79 acetylation and how this might be related to host shut-off. Reviews should help point the field to major unresolved questions.  

We thank the reviewer for this suggestion and have rewritten the conclusions section to include this information and renamed the section conclusions and outlook. The new section can be found at lines 395-434 and reads:

“Conclusions and outlook

The pathogenesis of IAV infections is a complex process that involves multiple cellular, viral and, occasionally, bacterial factors. Exposure to IAV can lead to a wide range of disease outcomes, and the host innate immune response is here a mixed blessing: it is indispensable for restricting viral replication and spread, but an overstimulation of the inflammatory response may also lead to severe disease. In efforts to explain pathogenic differences between IAV strains, mutations in IAV proteins have been identified that alter the receptor binding specificity, pH sensitivity, or function of IAV accessory proteins, such as NS1, PA-X and PB1-F2. These accessory proteins play a role in modulating the host response upon IAV infection and mutants derived from avian-adapted IAV strains may be less efficient at modulating the innate immune response in humans than mammalian-adapted IAV strains. IAV proteins or IAV infections have also been found to induce alterations in the infected host cell, such as H3K79 acetylation, release of nuclear RNAs, or Pol II read-through. How all these factors and modulations come together and add up to differences in virulence or pathogenicity is not fully understood.

In addition to mutations in the viral proteins, it has been discovered that aberrant IAV RNAs produced by erroneous polymerase activity can induce RIG-I activation and the expression of pro-inflammatory cytokines [21, 116]. Many questions remain regarding this role of aberrant IAV RNAs. For instance, it is presently unclear why avian-adapted IAV RNA polymerases appear to produce higher levels of mvRNAs than mammalian-adapted IAV RNA polymerases. In addition, it has not been explored if different sequences within aberrant RNA populations contribute equally to RIG-I activation or if sequence-dependent RIG-I activation differences exist among different aberrant RNAs. It is tempting to speculate, for instance, that IAV normally generates low levels of some aberrant molecules that induce no RIG-I activation, so they can act as RIG-I traps. Other work suggests that ZBP1 may also bind aberrant viral RNAs and induce necrosis [103, 104]. It is unclear at present if ZBP1 can only bind a sub-population of these aberrant RNAs since it only recognizes Z-form dsRNA, or if all IAV RNA species contain Z-form dsRNA and contribute to ZBP1 signalling.

In conclusion, the induction of a cytokine storm that underlies the pathogenesis of some human IAV infections, may be the result of a combination of a strong RIG-I activation and the inability of the virus to inhibit or evade key players of the host defence system. Indeed, passaging of wildtype IAV in non-IFN-responsive cells selected for IFN-inducing mutations at conserved positions in the IAV genome, and not just known sites in e.g., NS1. This view suggests that the entire IAV genome is optimized for both viral replication as well as counteracting or evading the host response. From that assumption it follows that due to suboptimal interactions with mammalian host factors, avian-adapted IAV proteins may be more likely to generate aberrant RNA products and be less efficient at suppressing the innate immune response. Future studies will be needed to better understand these interactions and their contribution to IAV virulence and pathogenicity.”

Some other minor points

the sentence in line 388 beginning "Indeed.." should be broken up or modified for clarity.

We agree with the reviewer that was an overly long sentence. We have changed this now as part of our response to the reviewer’s point about our conclusions (see above).

Fig 1B, the RNA strand is difficult to see in this color schema.

We have made the line thickness of the RNA strand thicker, so it stands out better.

Fig. 2 legend should indicate that viral polymerase components are colored blue.

We have corrected this in the legend to Fig. 2 (line 712-715). In addition, we have added this to the legend of Fig. 3

Fig. 3 legend, line 681-682 needs to be reworked.

We have amended the legend of Fig 3, lines 719-722, to:

“The IAV RNA polymerase produces four types of RNA: full length vRNA or cRNA segments, DI RNAs, mvRNAs, and svRNAs. Three of these RNA species contain partially dsRNA sequences and a 5′ triphosphate. When this dsRNA element is exposed to the solvent it can be bound by RIG-I. The viral proteins PB1, PB2 and PA are coloured bright blue, dark blue and grey blue, respectively.”